# Management of Rheumatoid Arthritis in Primary Care: A Scoping Review

**DOI:** 10.3390/ijerph21060662

**Published:** 2024-05-22

**Authors:** Francesco Inchingolo, Angelo Michele Inchingolo, Maria Celeste Fatone, Pasquale Avantario, Gaetano Del Vecchio, Carmela Pezzolla, Antonio Mancini, Francesco Galante, Andrea Palermo, Alessio Danilo Inchingolo, Gianna Dipalma

**Affiliations:** 1Department of Interdisciplinary Medicine, University of Bari “Aldo Moro”, 70124 Bari, Italy; a.inchingolo3@studenti.uniba.it (A.M.I.); pasquale.avantario@uniba.it (P.A.); gaetano.delvecchio@uniba.it (G.D.V.); carmela.pezzolla@uniba.it (C.P.); dr.antonio.mancini@gmail.com (A.M.); a.inchingolo1@studenti.uniba.it (A.D.I.); giannadipalma@tiscali.it (G.D.); 2PTA Trani-ASL BT, Viale Padre Pio, 76125 Trani, Italy; francesco.galante@aslbat.it; 3College of Medicine and Dentistry, Birmingham B4 6BN, UK

**Keywords:** cardiovascular prevention, chronic diseases, clinical audit, diagnosis, digital health, general practitioners, health costs, primary care, rheumatoid arthritis

## Abstract

Rheumatoid arthritis (RA) can lead to severe joint impairment and chronic disability. Primary care (PC), provided by general practitioners (GPs), is the first level of contact for the population with the healthcare system. The aim of this scoping review was to analyze the approach to RA in the PC setting. PubMed, Scopus, and Web of Science were searched using the MESH terms “rheumatoid arthritis” and “primary care” from 2013 to 2023. The search strategy followed the PRISMA-ScR guidelines. The 61 articles selected were analyzed qualitatively in a table and discussed in two sections, namely criticisms and strategies for the management of RA in PC. The main critical issues in the management of RA in PC are the following: difficulty and delay in diagnosis, in accessing rheumatological care, and in using DMARDs by GPs; ineffective communication between GPs and specialists; poor patient education; lack of cardiovascular prevention; and increase in healthcare costs. To overcome these criticisms, several management strategies have been identified, namely early diagnosis of RA, quick access to rheumatology care, effective communication between GPs and specialists, active patient involvement, screening for risk factors and comorbidities, clinical audit, interdisciplinary patient management, digital health, and cost analysis. PC appears to be the ideal healthcare setting to reduce the morbidity and mortality of chronic disease, including RA, if a widespread change in GPs’ approach to the disease and patients is mandatory.

## 1. Introduction

Rheumatoid arthritis (RA), the most common chronic rheumatic disease, is an immune-mediated chronic synovitis which not only leads to joint damage but can involve other organs such as the heart, kidneys, lungs, digestive system, eyes, skin, and nervous system [1,2,3,4,5,6,7]. The prevalence of RA can vary from 0.5% to 1% with a female-to-male ratio equal to 3:1 [8]. RA is associated with an elevated risk of disability, morbidity, and mortality [9,10,11,12]. Cardiovascular diseases (CVDs) are the most frequent RA comorbidities [13,14]. Despite the increasing understanding of pathogenetic mechanisms, which involve various macrophage cytokines (specifically interleukins IL-1, IL-17, IL-6, IL-8, IL-4, IL-5, IL-10, IL-12, IL-13, IL-35, IL-21, IL-22, TNF-alfa, TGF-beta, INF-gamma, RANKL, and GM-CSF), antibodies released by B lymphocytes and the presence of T helper lymphocytes (CD4+), and their interaction with traditional cardiovascular risk factors, the optimal strategies for the risk assessment, prevention, and treatment of RA are still unclear [15,16,17,18,19,20,21,22] (Figure 1).

The introduction of targeted biological and synthetic disease-modifying antirheumatic drugs (DMARDs) has improved clinical outcomes, reconfiguring the traditional cost composition of RA [23,24,25,26,27].

The yearly healthcare cost for RA, calculated by dividing lifetime healthcare expenditures by life expectancy, is USD 3812 [28].

Primary care (PC), provided by general practitioners (GPs), is the first level of contact for the population with the healthcare system [29,30,31]. The growing aging population and the related challenges such as chronic diseases and loss of autonomy have increased the demand for PC worldwide [32,33,34]. Increasing investment in PC is a key catalyst for overall healthcare cost reduction and population health improvement [35,36,37,38]. Figure 2 represents the organization of PC in European healthcare systems [39,40].

It has been established that the earlier the diagnosis of RA occurs and, consequently, the earlier therapy is undertaken, the faster clinical remission is achieved [41,42]. Consequently, the main tasks of GPs for the management of RA are early detection of the disease, fast referral to a specialist, first-line treatment, and the prevention of complications [43,44,45,46,47,48]. The new updated guidelines from the European League Against Rheumatism (EULAR) emphasize the importance of cardiovascular disease risk assessment at least once every 5 years for all RA patients [49,50]. Nevertheless, prevention of RA complications is only marginally applied in clinical practice, both in secondary and PC [51,52,53]. In addition, several gaps in the management of RA are well known, such as the significant gap between the availability of rheumatologists and the demand for rheumatology care, the rheumatologist shortage, regional disparities concerning the allocation of resources, and the lack of workforce planning [54].

The role of GPs in the diagnosis and management of RA can vary considerably at a regional level, influenced by several factors such as training, available resources, and health policies. Regional differences may influence the quality and timeliness of access to rheumatology care [55]. For example, in some areas with shortages of rheumatology specialists or with long waiting lists for specialist visits, GPs could take a more active role in the long-term management of patients with RA, monitoring responses to treatment and making therapeutic adjustments in collaboration with rheumatologists. Disparities in access to specialist care may also influence GPs’ approach to managing RA [56,57]. In regions with limited resources or low availability of advanced biologic treatments, GPs may have to rely primarily on conventional therapies and closer monitoring of patients to manage the disease optimally. On the other hand, in areas with greater resources and access to innovative therapies, GPs may have greater confidence in referring patients for more aggressive and personalized treatments managed by specialists [58]. In many regions, the shortage of GPs trained in the early identification and management of rheumatic diseases can delay the diagnosis and treatment of RA, leading to increased joint damage and disability in patients [59]. Investing in primary care through specialized training, evidence-based decision support, and access to appropriate diagnostic resources can help improve the efficiency and effectiveness of RA management [60].

The recent COVID-19 pandemic has accelerated the adoption of digital health (eHealth) technologies, paving the way for greater incorporation of these technologies into PC settings [49,61,62,63,64,65,66,67,68,69]. eHealth solutions can facilitate primary healthcare delivery and contribute to the realization of universal health coverage [70,71]. eHealth technologies include mobile devices, telemedicine, artificial intelligence, and big data analytics [72,73,74,75,76,77,78,79]. The application of eHealth tools in PC leads to improvements at various levels, e.g., prevention, diagnostics, therapy, patient management, homecare for the elderly, risk prediction, clinical decision support, accuracy of diagnosis, optimization of healthcare resources, patient satisfaction of healthcare, equity of care, and health cost reduction [61,80,81,82,83,84,85]. However, there are limitations to the use of eHealth, such as existing socio-cultural norms, geographic inequalities in access to technology, and disparities in digital literacy [70,86,87,88].

During the SARS-CoV-2 pandemic, there has been an increase in the use of eHealth solutions in rheumatology to ensure the continuity of care and the safety of patients and caregivers [82,89,90]. The retrospective study by Avouac J. et al. demonstrated the effectiveness of telemedicine in the management of RA by rheumatologists [91]. In a Canadian study that analyzed patient perceptions, telemedicine even proved to be superior to face-to-face consultations in ensuring continuity of care and reducing stress, time, and costs [89]. A French work demonstrated that PC represents an ideal setting for the use of eHealth tools in RA patients, as it promotes therapeutic compliance [92]. Telemedicine is associated with significant improvements in RA patients in terms of disease activity, function, physical activity, and self-efficacy [93,94,95].

At the current state of knowledge, this is the first comprehensive review regarding the management of a chronic rheumatological disease in primary care settings. Specifically, this scoping review aims to collect the main critical issues of RA management and, in parallel, to identify strategies to overcome them, with particular attention on the application of eHealth solutions.

## 2. Materials and Methods

In consideration of the broad nature of the topic and of the purpose of this research, finalized to identify gaps in knowledge, we have chosen to process the results and describe them qualitatively in a scoping review [96]. The search was performed following the method of Arksey and O’Malley (2005) and the results were reported in accordance with the Preferred Reporting Items for Systematic Reviews and Meta-Analyses statement for reporting scoping reviews (PRISMA-ScR) [97,98].

### 2.1. PCC Framework

As recommended by Pollock et al., we applied the PCC (population, concept, and context) framework to identify the main concept for the primary question of this scoping review [99].

Primary review question: What are the critical issues and strategies in the management of rheumatoid arthritis in primary care settings?

Population/participants: adult RA patients;Concept: critical issues and strategies used to manage RA by GPs;Context: PC settings.

### 2.2. Sources of Information and Search Strategy

The electronic databases PubMed, Scopus, Web of Science, and Cochrane were systematically searched to find papers that matched our topic dating from 1 January 2013 to 30 October 2023. The Medical Subject Headings (MESH) terms entered in search engines were “rheumatoid arthritis” AND “primary care” (Table 1).

### 2.3. Inclusion and Exclusion Criteria

The inclusion criteria were the following: (1) human subjects; (2) English-language articles; (3) randomized controlled trials (RCTs), clinical trials, cohort studies, and longitudinal studies; (4) open access; (5) primary care settings.

The exclusion criteria were the following: (1) other languages except English; (2) reviews, case reports, and case series; (3) off-topic articles; (4) hospital settings or healthcare settings other than primary care.

### 2.4. Screening and Data Charting

After eliminating duplicates manually, the title, abstract, and full text were screened against the inclusion and exclusion criteria. The screening of records occurred in two phases: the first consisted of screening the title and abstract, and the second consisted of reviewing the complete text. The primary reviewer (M.C.F.) performed 100% of the screening, and two additional reviewers (A.M.I./A.D.I.) independently performed the screening by dividing the articles into two equal parts. Doubts were resolved through discussion with the senior reviewer (F.I.). The electronic search results were exported to an Excel spreadsheet. Data graphing was primarily conducted by M.C.F. and controlled by G.D. No quality assessment was performed for this scoping review as the aim was to summarize the evidence in the literature on the topic to inform future clinical practice, without including or excluding quality-based studies.

## 3. Results

### 3.1. Characteristics of Included Articles

A total of 400 records were identified using the keywords “rheumatoid arthritis” AND “primary care”. When applicable, the automatic filters entered were only human, only in English, only clinical studies, no reviews, and free full text. The consulted databases were PubMed (29), Scopus (270), Web of Science (37), and Cochrane (64).

The inclusion and exclusion criteria were applied based on the analysis of the title and the abstract. Only studies that focused on criticisms and strategies of RA management in PC were selected. Emphasis was placed on the application of eHealth processes in PC. Studies concerning the approach to RA in other health settings, such as tertiary care and hospitals, as well as dealing with other aspects of RA, were excluded because they were off-topic.

A total of 61 records were included in the final analysis (Figure 3), of which only 5 studies were RCTs.

### 3.2. Result Presentation

The extracted data will be presented in the form of tables to align with the objective and scope of this scoping review. A qualitative thematic analysis will be carried out to provide an overview of the literature on the topic. The results will be discussed based on practice and research and a useful conclusion will be drawn for future studies in this field (Table 2).

## 4. Discussion

The discussion of this review is made up of two parts: the first concerns the critical issues in the management of RA in CPs, and the second focuses on the solutions to bypass them (Figure 4).

### 4.1. Criticisms of Rheumatoid Arthritis (RA) Management in Primary Care (PC)

#### 4.1.1. Delay and Difficulty in the Diagnosis of Rheumatoid Arthritis (RA)

The first critical element in the management of RA in CP is the difficulty and delay by GPs in making a diagnosis [161,162,163]. Early diagnosis is fundamental for achieving remission in the initial phase of RA [164,165,166]. The study led by van Beers-Tas et al. identified a crucial temporal pattern in PC for individuals later diagnosed with inflammatory arthritis (IA). The authors revealed a marked rise in visits for musculoskeletal symptoms and infections almost one and a half years prior to the official diagnosis. These findings underscore the importance of recognizing these early indicators, suggesting that interventions or heightened awareness within the healthcare system could lead to earlier detection and management of IA [100]. In a similar study concerning 4549 adults visiting PC settings, Hider et al. found that a substantial proportion (54.3%) of patients reported non-specific musculoskeletal problems, such as joint pain, stiffness, and swelling. Notably, these symptoms were commonly presented by individuals referred to PC for non-musculoskeletal issues. This work underlined the need for enhanced strategies for the early detection and management of IA in PC settings [101].

In parallel with the non-specificity of the symptoms reported by patients, the difficulty for GPs in interpreting the signs and symptoms of inflammatory-type pain also contributes to RA diagnostic delay. Newsum et al. revealed that GPs rely primarily on the classic signs of inflammation for diagnosis. Critical elements evaluated routinely in secondary care, such as morning stiffness, compression testing, and family history, were rarely recorded by family doctors. This emphasizes the need for better education, awareness, and application of guidelines for GPs to identify and manage IA [102].

Another critical issue that prevents an easy diagnosis of RA is the poor and incorrect use of diagnostic tests by GPs [167,168,169]. Morsley et al. scrutinized the use of RF tests in Spanish PC. Incorrect utilization of RF testing not only leads to delayed diagnosis but also incurs increased costs. Considering that 10% of patients tested for RF did not have an RA diagnosis, the authors recommended limiting testing to individuals with higher pre-test probability, advocating for a more targeted approach to reduce health costs [103]. Examining 173 RA patients, Singh et al. revealed a significant gap in the utilization of rheumatologic testing by GPs before referring patients with arthritis. ACPA and RF were prescribed in only 34.45% of cases. Notably, almost half of the patients experienced a delay of at least one year before referral [104].

Miller et al. evaluated the diagnostic accuracy of RF testing for RA in PC settings and its impact on referral times. Analyzing data from the Clinical Practice Research Datalink, they found that of 62,436 RF tests, 7.5% were positive, with 57.8% of incident RA cases being seropositive. RF exhibited a positive likelihood ratio of 9.5 and a negative likelihood ratio of 0.5, with a sensitivity and specificity of 57.8% and 93.9%, respectively. However, the median time to referral after a positive RF test was significantly shorter than for negative tests (54 vs. 150 days). The study suggests that RF may aid RA diagnosis but performs poorly in excluding it and may delay referral. However, the authors did not find an association between ACPA utilization and the delay in treatment initiation. The key recommendation is to encourage GPs to adopt earlier diagnostic tests, especially ACPA, for suspected RA cases, potentially reducing delays in diagnosis [105].

#### 4.1.2. Delay in the Referral of Rheumatoid Arthritis (RA) Patients to Secondary Care

The study by Saad et al. introduced another concerning factor: the delay in the referral of RA patients to rheumatologists. The study emphasized the need for family physicians to be well versed in the guidelines, such as the European Alliance of Associations for Rheumatology (EULAR) 2016 guidelines for RA, to expedite referrals and improve patient outcomes [106]. Likewise, Ledingham et al. assessed if patients with newly diagnosed inflammatory arthritis (IA) met the National Institute for Health and Care Excellence (NICE) quality guidelines. The results showed a considerable gap between the standards and the delivery of care, subject to geographic variability, underscoring the need for systemic improvements [107]. Further complicating the pathway to rheumatology consultation, the study by Stack et al. conducted in the UK revealed alarming delays at different stages. The median time between symptom onset and seeing a rheumatologist was 27.2 weeks, with substantial contributions from patient delay, GP delay, and hospital delay. The authors emphasized the necessity for a thorough plan to speed up the entire procedure [108].

Examining access to rheumatologists in Ontario, Canada, the study by Widdifield et al. assessed the percentage of RA patients seen by a rheumatologist within 3, 6, and 12 months of suspected diagnosis. Despite improvements over time, disparities persisted, influenced by external factors such as proximity to rheumatologists, socioeconomic status, and male physician sex. These findings highlight the need for addressing geographical and socioeconomic barriers to ensure timely access to rheumatological care [109]. By studying access to rheumatologists for patients with arthritis and AI in Ontario, Badley et al.’s study identified barriers to access, particularly for populations with low access to GPs and patients in need of rheumatological care. The study emphasized the importance of overcoming these barriers to ensure optimal care for patients with arthritis, stressing the crucial role of PC access in facilitating specialized rheumatic care [110].

The delay in referral to specialist care is a source of great dissatisfaction for RA patients, as Chilton et al. found through the administration of semi-structured interviews. The main difficulties reported by survey participants are as follows: “family persuasion”, “lack of continuity in care”, “push for postponement”, “strained relationships”, and “wasted time”. Moreover, there are three key factors on which the specialist diagnosis depends: (i) patient factor: the time it takes for a patient to consult a GP; (ii) PC factor: the time it takes for GPs to refer their patient to a specialist; (iii) secondary care factor: the time it takes for a patient to be seen by a specialist after referral. The second factor is the one that has the greatest impact on the diagnostic delay. From the analysis of the interviews, the authors concluded that patients with RA and their families experience the frustration of not feeling heard, believed, or understood and have associated these feelings with their delay in being referred for second-level care by GPs. Therefore, GP awareness campaigns on the physical and psychological needs of RA patients would be useful [111].

Instead, the study conducted by Laires et al. concerned stakeholders’ perspectives in Portugal, elucidating problems related to accessibility, diagnostic challenges, and inefficient referrals. The proposed solutions included optimizing national registries, distributing information on rheumatic symptoms in PC, and promoting collaboration between rheumatologists and GPs [112].

#### 4.1.3. Difficult in the Management of Rheumatoid Arthritis (RA) Treatments

Lee et al. revealed that, despite American College of Rheumatology (ACR) guidelines recommending earlier and more aggressive treatment, several older adult RA patients are not treated with DMARDs, including biologic DMARDs. This difficulty is shared both by rheumatologists and GPs, who use DMARDs in 56% and 30% of patients, respectively. Only 74% of patients are visited by a rheumatologist, signaling a need for increased awareness and adherence to treatment guidelines among healthcare providers. On the other hand, a high prescription rate of glucocorticoids (GCs) is observed among RA patients, especially from GPs [113]. Looking back at a large cohort of patients, Black RJ et al. found that in the United Kingdom, about 50% of patients initially diagnosed with RA took GCs prescribed by GPs, at an average dose of 7.5 mg/day. Factors predisposing patients to treatment with GCs are advanced age and CVD [114]. Non-steroidal anti-inflammatory drugs/cyclooxygenase 2 inhibitors are the other drugs most ordered by GPs according to the observational study by Widdifield J. et al. [115]. Finally, patients taking MTX, the first-line conventional DMARDs for RA, are subjected to laboratory monitoring (e.g., blood count and transaminase) more frequently than 3 months, as recommended by guidelines of the British Society of Rheumatology (BSR). The over-prescription of laboratory tests is correlated with poor education, both of physicians and patients, and increased health costs [116].

#### 4.1.4. Ineffective Communication between Rheumatoid Arthritis (RA) Healthcare Professionals

RA management in PC is affected by ineffective communication between healthcare professionals, who in this specific case are rheumatologists and GPs [170,171,172]. The study by Wong et al. exposed shortcomings in the quality and continuity of information transfer. Referral letters are deficient in critical information, impairing the coordination and continuity of care for RA patients. For this purpose, the authors highlighted the need for having open lines of contact [117].

#### 4.1.5. Lack of Cardiovascular Disease (CVD) Preventive Care

A major shortcoming is the lack of CVD preventive care for RA [173]. For this purpose, Barber et al. assessed the performance of CVD quality indicators. The authors revealed prevalent CVD risk factors, including hypertension, obesity, smoking, and dyslipidemia. The study suggested a need for improved coordination and communication between rheumatology and PC to address cardiovascular risks effectively in RA patients [118]. The study by Bartels et al. investigated the challenges in providing preventive care for CVD in RA patients. Preventive treatment is typically inadequate, despite the elevated risk of cardiovascular disease. According to interviews with patients, rheumatologists, and PC physicians, receiving preventative treatment requires recognizing and addressing risk factors that are often overlooked. The study suggested improving recommendations for GPs, increasing patient engagement, and introducing system-based RA-CVD prevention Bartels et al. delved into the intriguing relationship between RA and the probability of receiving a diagnosis of hypertension. Among patients with undiagnosed hypertension, those with RA had a lower probability (36%) of receiving a diagnosis of hypertension than those without RA (51%). Despite having more total visits, RA patients were less likely to be diagnosed. The authors highlighted potential gaps in recognizing and managing comorbidities in RA patients, prompting increased awareness among healthcare professionals [119].

#### 4.1.6. Increased Health Costs for Rheumatoid Arthritis (RA)

Focusing on the long-term economic impact of RA management, the study by Nikiphorou et al. explored medical costs associated with IA a decade after disease onset. The study revealed a substantial increase in direct medical costs over a 6-month period, primarily driven by the use of biologic DMARDs for the RA subgroup (51% of total costs). The findings underscore the evolution of health costs related to RA, which tend to increase in direct proportion to the progression of the disease [121].

Herein, the studies examined collectively emphasize the urgency of addressing the multifaceted challenges in the diagnosis and management of RA. By fostering collaboration, optimizing diagnostic strategies, and enhancing communication, healthcare professionals can bridge the existing gaps and pave the way for a more efficient and patient-centered rheumatological care system.

### 4.2. Strategies of Rheumatoid Arthritis (RA) Management in Primary Care (PC)

#### 4.2.1. Early and Accurate Diagnosis of Rheumatoid Arthritis (RA)

The improvement of anamnesis has been recognized as a crucial factor in addressing the challenges faced by GPs in diagnosing and referring patients with suspected RA [174]. Despite national guidelines recommending prompt referrals within three days of persistent synovitis presentation, the study by Scott C. et al. revealed that a minority of GPs adhere to this timeline [175]. The overreliance on investigations, such as rheumatoid factor (RF) testing, is identified as a modifiable barrier to early referral. The authors emphasized the significance of clinical history and examination findings in the early diagnosis of RA, especially small joint swelling and pain, and advocated for a paradigm shift in GPs’ approaches, prioritizing clinical findings over investigations to meet referral timelines, ultimately aiming to enhance the timely diagnosis and treatment of RA for improved patient outcomes [122].

Van Dijk B.T. et al. focused on the discriminative ability of self-reported functional impairments in early detection among patients with suspected but uncertain IA. Utilizing data from two Early Arthritis Recognition Clinics in the Netherlands, the Health Assessment Questionnaire Disability Index (HAQ) was employed to measure physical functioning. The authors suggested that a simple question about difficulties with dressing could aid in early IA detection, providing a valuable tool for healthcare professionals, particularly those with limited experience in joint examination. This approach aligns with the need for a more patient-centric approach in the early diagnosis of RA [123].

Garcia-Montoya’s study concerned the association between the risk of progression to IA in individuals with new musculoskeletal (MSK) symptoms. Focusing on anti-cyclic citrullinated peptide (anti-CCP) assays, the study identified specific predictors, such as high anti-CCP levels and pain in the hands or feet, indicating an increased risk of progression to IA. The study underlined the potential of routinely available tests and joint symptoms in discriminating and prioritizing referrals to rheumatology, avoiding delayed diagnoses [124].

Nam J. et al. further reinforced the predictive value of biomarkers in RA. By selecting individuals with new non-specific MSK symptoms without clinical synovitis, the authors enriched the prevalence of anti-CCP positivity, highlighting a high risk of rapidly developing RA in this cohort. This selective case approach suggests potential improvements in the clinical relevance of anti-CCP antibodies in predicting the progression to IA [125].

Morsley and colleagues underscored the critical assessment of diagnostic test efficacy and cost-effectiveness, focusing on RF testing for RA in PC. Among 495,434 tested patients, only 2.6% of RF-positive cases were diagnosed with RA within a year. With a sensitivity of 70.8%, specificity of 78.7%, and a substantial cost of approximately EUR 3,963,472, equivalent to EUR 1432 per true positive, the authors recommended the need for more precise patient selection for RF testing to reduce costs in primary practice [103].

Almoallim H.’s contribution focused on providing standardized and validated criteria for early referral to a rheumatologist. Analyzing various demographic variables, patient-reported complaints, physical examination results, and biomarkers, the authors identified nine variables with high specificity and good predictive value. These variables, including loss of appetite, joint swelling, positive tests for RA, and anti-CCP antibodies, are recommended as early referral criteria. This approach aims to assist GPs in making appropriate and timely referrals for suspected IA [126].

Van Delft Etam et al. explored the application of criteria for early referral of RA patients. The study assessed the diagnostic performance and clinical utility of the Rotterdam Early Arthritis Cohort (REACH) and the Clinical Arthritis Rule (CARE) referral rules in an independent population of unselected patients from PC. Both validated rules show a net benefit in recognizing IA and inflammatory rheumatic diseases compared to usual care, with CARE demonstrating superiority over REACH. The authors also introduced a composite referral rule for inflammatory rheumatic diseases, incorporating various parameters such as sex, age, joint features, and duration of symptoms. This composite rule exhibits increased diagnostic performance, suggesting its potential applicability in PC settings [127].

Ten Brinck RM et al. dealt with the uncertainties of GPs in detecting synovitis through joint examination. By developing and validating a rule composed of clinical characteristics, the study aimed to assist GPs in identifying IA when in doubt. The simplified rule, consisting of seven items, demonstrated reasonable discriminative ability for IA. This approach contributes to decision-making in patients with suspected IA, potentially increasing the appropriateness of healthcare utilization [128].

Nielen M. et al. focused on the validity of IA diagnosis in PC based on electronic medical records (EMRs). By systematically reviewing EMRs, the study confirmed the diagnosis of IA in 70.8% of patients with a diagnostic code of IA in the database. The proposed algorithm, incorporating additional diagnostic information, increased the percentage of correctly diagnosed IA patients to 78%. This algorithm aims to improve the diagnostic validity of IA in PC settings [129]. Similarly, Muller S. et al. proposed an updated algorithm for identifying RA in PC using the Clinical Practice Research Datalink (CPRD). The updated algorithm, reflecting changes in coding practices and the introduction of new treatments, revealed an increased rate of confirmed RA diagnoses (86%). The authors recommend the updated algorithm for future studies in the field, emphasizing its potential to enhance the reliability of epidemiological studies and CAs [130].

Cummins et al. assessed the application of the 2010 American College of Rheumatology (ACR)/European League Against Rheumatism (EULAR) criteria for the early diagnosis of RA in PC. The prospective application of these criteria improves triage decisions and reduces waiting times for rheumatologist diagnosis. This strategy demonstrates high sensitivity and specificity, with a significant reduction in median wait times for referrals fulfilling the triage criteria. Implementing these criteria is a priority in PC and can lead to earlier diagnosis and treatment for RA patients, particularly in areas with a scarcity of rheumatologists [131].

In conclusion, the reviewed studies collectively highlight the importance of a comprehensive and patient-centric approach in the early diagnosis of RA. From improving anamnesis and clinical examination to leveraging biomarkers and validated referral criteria, it is possible to enhance the diagnostic process, reduce delays, and ultimately improve patient outcomes. The application of these findings into PC practices holds the potential to reshape the landscape of RA diagnosis and management.

#### 4.2.2. Rapid Access to Specialist Rheumatological Care

RA is associated with an increased risk of comorbidities, such as CVD, pulmonary diseases, diabetes, and depression [15,176,177,178,179]. Nevertheless, comprehensive recommendations for managing these related conditions are lacking [180]. In this regard, Kvien et al., based on over 180 interviews with specialists from 12 European rheumatology centers, highlighted the challenges faced in RA management, namely delays in diagnosis, shortages of rheumatologists, and a lack of awareness among GPs. These obstacles not only hinder the timely initiation of treatment but also contribute to the worsening of associated comorbidities. The authors also suggested 18 best-practice interventions according to the three phases of the disease, namely suspected RA, recent diagnosis of RA, and established RA. Among the most notable interventions, support for self-management, early arthritis clinics, rapid access to care through online referral, triage, and ultrasound-guided diagnosis are included. In addition, crucial components of an effective intervention strategy are the following: the introduction of specialists dedicated to comorbidities, improved communication with primary care, and the integration of patient registries into clinical practice. The proposed intervention guide will improve speed and efficiency in accessing specialized care, ensuring that patients receive the support they need at every stage of the disease [132].

#### 4.2.3. Effective Communication among General Practitioners (GPs), Rheumatologists, and Patients

Effective communication among GPs, rheumatologists, and patients is crucial for enhancing the management of RA comorbidities, especially CVD [181,182,183]. Barber C.E. et al. assessed the performance of 11 quality indicators for CVD management in RA patients, revealing significant variations in the management of CVD risk factors, with areas of low performance such as inadequate documentation of a formal cardiovascular risk assessment, failure to communicate increased cardiovascular risk to GPs, and a lack of discussion on the risks and benefits of anti-inflammatory agents in patients with CVD risk. These findings underscore the need for improvements in CVD risk management, with particular attention to communication between rheumatologists and GPs [118]. On the other hand, Kvien et al. prepared interviews with specialists in 12 European rheumatological centers. The authors identified challenges along the patient’s journey and proposed best practices to enhance the care of patients with RA and comorbidities. Enabling self-management, establishing clinics for early arthritis, ensuring rapid access to care, improving communication with general medicine, and integrating patient registries into daily clinical practice are included in the authors’ recommendations [132].

Russell-Westhead M. et al. examined the effectiveness of group consultations in treating IA. Their results suggested that group consultations can be sustainable, clinically effective, and efficient for the monthly review of active disease and annual review of stable disease. Key success factors include efficiency, empathy, education, patient involvement, and autonomy [133]. Furthermore, Laires P.A. et al. investigated barriers to access to specialist care for RA patients in Portugal. Identified barriers are the following: difficulty in accessing PC services, delays in diagnosis by GPs, inefficiencies in referrals to specialized care, and controlled processes in prescribing biologic drugs in public hospitals. Proposed solutions include improving epidemiological and clinical knowledge of RA, enhancing disease understanding among patients and GPs, and increasing awareness of current treatment guidelines [112].

Bartels C.M. et al. examined the prevention of CVD in RA patients. The authors revealed gaps in CVD prevention, underlining the importance of identifying and addressing CVD risk factors. The barriers include the lack of awareness of CVD risk by patients and GPs, the absence of systematic identification of risk factors by rheumatologists, and challenges related to communication and CVD risk management [120].

Finally, Magliah M. et al. dealt with the agreement between rheumatologists and GPs trained by rheumatologists in the early diagnosis of IA. The results indicate promising agreement in MSK examinations performed by GPs compared to rheumatologists, suggesting that standardized training could improve the accuracy of early diagnosis of IA [134]. Fonseca et al. investigated the impact of a referral support program (RSA) on identifying patients with RA and axial spondylarthritis (SpA). In this RCT study, 28 PC units were randomly assigned to RSA or control groups. Both groups referred suspected RA or SpA patients to the rheumatology unit of a reference hospital, aiming to evaluate accurate diagnosis by rheumatologists. The results showed a positive trend for RSA, particularly in diagnosing any type of arthritis in the RA subgroup, though the effect was moderate and not statistically significant. In the SpA subgroup, RSA showed a positive effect but lacked statistical significance. The study emphasized the importance of effective communication among GPs, rheumatologists, and patients in implementing referral programs to enhance early diagnosis and management of RA and SpA [135].

In conclusion, effective communication among GPs, rheumatologists, and patients could enhance the management of comorbidities associated with RA, with particular attention on CVD prevention and active patient involvement.

#### 4.2.4. Screening of Risk Factors and Comorbidities

The effective management of rheumatic diseases such as RA extends beyond the control of joint symptoms to include the prevention of comorbidities, particularly CVD, which pose a significant threat to RA patients [46,184,185].

Movahedi M. et al. conducted a consistent cohort study to assess the risk of developing diabetes mellitus (DM) related to the use of glucocorticoids (GCs) in RA patients. Involving 21,962 patients in the United Kingdom (Clinical Practice Research Datalink [CPRD]) and 12,657 in the United States (National Data Bank for Rheumatic Diseases [NDB]), the study indicated that the use of GCs represents a significant independent risk factor for the development of DM in RA patients, highlighting the importance of dosage and treatment duration. The hazard ratios (HRs) were 1.30 (CPRD) and 1.61 (NDB), respectively. Therefore, GC administration in RA patients should be carefully considered [136]. Stouten et al. focused on comparing the incidence of comorbidities and the prescription of pain relievers in RA patients, psoriatic arthritis (PsA), or SpA compared to controls. Using data from the Intego registry in Flanders, Belgium, the study revealed an increased burden of comorbidities in patients with chronic inflammatory rheumatic conditions, especially RA and PsA. Depression, cardiovascular diseases, and opioid use were particularly high in patients with RA and PsA, revealing the need for targeted therapeutic interventions [137].

Monk et al. investigated the implementation of CVD screening guidelines in PC for RA patients. Analyzing a database of PC consultations in the United Kingdom, the study revealed that—despite a comparable excess risk of CVD in RA and diabetes—additional screenings for CVD are not organized in PC settings. This assumption suggests the need for greater commitment to active CVD screening in RA patients [138].

Nikiphorou et al. focused their attention on the relationship between RA and CVD. They conducted a retrospective case–control study to assess the burden of CVD in early RA. Identifying an excess of stroke and heart failure before the diagnosis of RA, the study highlighted an increased risk of CVD post-diagnosis, emphasizing the need for comprehensive management strategies to address this risk in patients with early RA [139].

In two studies published in 2016 and 2017, respectively, Joseph et al. contributed valuable insights into the association between smoking and mortality in RA patients. The studies utilized the Clinical Practice Research Datalink to identify an incident cohort of RA patients. Current smoking was associated with an increased risk of all-cause mortality, CVD, and respiratory tract infections. Importantly, each year of smoking cessation was associated with a reduced risk of mortality, highlighting the potential benefits of smoking cessation programs for RA patients [140,141].

Abhishek et al. extrapolated data from the Clinical Practice Research Datalink to analyze temporal trends in all-cause and cause-specific mortality in RA. The cohort comprised 21,622 incident RA cases and 86,488 matched controls, followed for up to 5 years. Mortality rates for RA cases and controls were 26.90 and 18.92 per 1000 person-years. From 2005 to 2009, the RA mortality rate decreased by 7.7% annually, while controls improved by 2.2% annually from 1990 to 2009. Despite a 32% excess mortality risk in the entire cohort, cases incident after 2006 exhibited a 15% decrease. The authors observed that improvements in RA management may contribute to reducing RA mortality, but challenges persist in addressing associated health risks [142].

In conclusion, these studies collectively contribute valuable insights into the multifaceted aspects of RA, ranging from the impact of GCs and comorbidity burdens to the association between smoking, CVD, and mortality. Understanding these factors is crucial for developing targeted interventions and improving the overall management of RA patients.

#### 4.2.5. Clinical Audit (CA)

A CA emerges as a fundamental strategy in clinical governance, representing a crucial tool for aligning guidelines with clinical practice [186,187]. It is an initiative conducted by healthcare professionals such as GPs, nurses, and other healthcare practitioners, which aims to enhance the quality and the outcomes of care through a structured and systematic peer review [188,189]. The goal is to scrutinize clinical activities considering explicit standards and to provide targeted changes, if necessary, by assessing the appropriateness of processes based on the best available evidence [190,191].

Two articles, resulting from a comprehensive national CA of rheumatology units in England and Wales, provided significant data on the early management of the disease. In the study by Pratt A.G. et al., the national CA involved over 6000 patients with suspected IA, focusing on measuring the performance of rheumatology services against the quality standards of the National Institute for Health and Care Excellence (NICE). The results revealed significant discrepancies between NICE standards and actual practice. The PC physicians referred only 17% of patients within 3 days of the first presentation, and only 38% received a specialist evaluation within 3 weeks of referral. Only 53% of RA patients were treated with DMARDs within 6 weeks of referral. Additionally, the CA highlighted a lack of uniformity in providing structured information and educating patients about their condition. Only 59% received structured education on arthritis within 1 month of diagnosis, and only 27% achieved the agreed therapeutic target within 3 months of specialist review. A crucial component of the CA was the detection of significant geographic variability in the management of RA. This underlines the need for a more uniform and coordinated approach to delivering PC nationally [143].

Concurrently, Stack R.J. et al. analyzed the impact of treatment time based on autoantibodies in RA patients. The results highlighted delays in presentation to GPs for patients with specific autoantibodies, suggesting the importance of considering individual characteristics in the early management of the disease. Both studies revealed that CA emerges as an indispensable tool in the management of RA in PC. Through detailed analysis of collected data, CA identify critical areas requiring immediate improvement. CA, capturing the voices and experiences of patients, become an essential tool for understanding specific challenges, both at the individual and geographic levels [144].

In conclusion, CAs bridge the gaps between quality standards and actual practice. Through the implementation of corrective measures based on CA findings, it is possible to promote more timely, uniform, and personalized disease management, contributing to improving the quality of life for RA patients.

#### 4.2.6. Active RA Patient Involvement

Persell and colleagues demonstrated that encouraging a proactive approach involving patient engagement in the management of cardiovascular risk factors led to significant improvements in RA. During the six-month intervention period, an increase in the use of statins for primary prevention was observed, rising from 18.4% to 23.8%, while diagnoses and control of hypertension significantly improved [145].

Despite the advent of biologic therapies, methotrexate (MTX) remains the first-choice DMARD for treating RA [192]. However, non-adherence to MTX can compromise treatment effectiveness. Curtis and colleagues developed and validated the Methotrexate Experience Questionnaire (MEQ) to identify and characterize non-adherence to MTX. Through a retrospective cross-sectional study conducted in the United Kingdom, it was found that the MEQ is a reliable and valid tool for assessing the adherence of RA patients to MTX treatment, offering deeper insights into individual reasons for non-adherence and facilitating clinician–patient discussions [146].

Moreover, it is crucial to consider the active involvement of patients in the process of redesigning healthcare systems, as highlighted by Lopatina et al. In a case study, Lopatina explored how patient voices were incorporated into the redesign process of an element of the healthcare system, namely the centralized intake system for referrals from primary care to rheumatologists for patients with suspected RA. Active patient involvement in activities such as project vision development, prospective data collection through patient-to-patient studies, and definition of future evaluation strategies proved to significantly contribute to creating a healthcare system more responsive to patient needs [147].

In conclusion, active patient involvement in managing cardiovascular risk factors, treatment adherence, and healthcare system redesign emerge as key elements in improving care and quality of life for RA patients. These approaches offer a comprehensive framework for addressing the complexity of RA management and its comorbidities, shifting the focus not only to joint symptoms but also to overall patient well-being.

#### 4.2.7. Interdisciplinary Management of RA

The interdisciplinary management of RA involves not only GPs and rheumatologists but also other healthcare professionals, such as nurses and physiotherapists [193,194,195,196].

Puchner et al. addressed the critical issue of optimizing patient flow for RA patients with the dual aim of enhancing health outcomes and reducing direct healthcare costs. The author underscored the importance of involving and coordinating various healthcare professionals, including rheumatologists, GPs, physiotherapists, nurses, and other specialists, to ensure a comprehensive and integrated approach to patient care. This interdisciplinary perspective is crucial for addressing the multifaceted nature of RA, which encompasses not only clinical aspects but also social, psychological, and economic dimensions. Moreover, the author emphasized that effective collaboration among different healthcare stakeholders can contribute to improving patient flow through the healthcare system, reducing waiting times for specialist appointments, diagnostic procedures, and treatments. This not only leads to better clinical outcomes but can also reduce direct healthcare costs by enabling the more efficient use of healthcare resources. The article also underscores the importance of actively involving patients in the decision-making process regarding their care pathway, promoting patient autonomy and active participation in managing their condition [148].

In an RCT, Bakker et al. focused on a comprehensive intervention for individuals with IA, particularly emphasizing the role of physiotherapists in leading a multimodal vocational program. The proposed intervention involves collaboration between various healthcare professionals, with physiotherapists playing a central role. RA affects multiple aspects of a person’s life, including physical function, work participation, and overall well-being, necessitating a holistic approach. Furthermore, the article outlines how the intervention integrates various components, such as exercise therapy, occupational therapy, and psychosocial support, to address the multifaceted challenges faced by individuals with IA in maintaining employment and participation in vocational activities. This multimodal approach acknowledges the interconnectedness of physical, psychological, and social factors in influencing work ability and aims to provide comprehensive support tailored to the individual’s needs. Overall, the article highlights the crucial role of interdisciplinarity in designing and implementing effective interventions for individuals with IA, particularly in the context of vocational rehabilitation. By integrating the expertise of physiotherapists, occupational therapists, vocational experts, and other healthcare professionals, the proposed intervention aims to improve outcomes and enhance the overall quality of life for individuals living with IA [149].

Similarly, Ndosi et al. provided evidence of non-inferiority regarding RA care provided by nurses working in PC. This RCT evaluated the role of nurses as primary caregivers, working collaboratively with other healthcare professionals, including rheumatologists, GPs, and allied health professionals. The authors highlighted the specific contributions of nurses in the community setting, where they play a central role in providing ongoing support, education, and monitoring to RA patients. Nurses are involved in various aspects of care, including medication management, symptom assessment, patient education, and psychosocial support. This multifaceted involvement underscores the importance of nurses as key members of the healthcare team, working alongside other disciplines to optimize patient outcomes. Moreover, the article evaluates the cost-effectiveness of nurse-led care in the community, emphasizing the potential economic benefits of this model of care. By delivering care in the community setting, nurses can help reduce the burden on hospitals and specialty clinics, improve access to care for patients, and potentially lower healthcare costs [150].

Likewise, in an RCT, Hider et al. investigated the feasibility of a nurse-led integrated care review for individuals with inflammatory rheumatological conditions in PC settings, highlighting the importance of interdisciplinarity in improving patient care. The study proposed a nurse-led model that integrates care across various healthcare disciplines, including primary care physicians, rheumatologists, nurses, and other allied health professionals. The authors underscored the pivotal role of nurses in leading the integrated care review process. Nurses are positioned as key coordinators of care, facilitating communication between patients and healthcare providers, conducting comprehensive assessments, providing patient education, and coordinating referrals to other healthcare professionals as needed. This highlights the importance of recognizing the expertise and contributions of nurses in the management of inflammatory rheumatological conditions within PC settings. Moreover, the study evaluated the feasibility and acceptability of nurse-led integrated care, highlighting the positive feedback from both patients and healthcare professionals. The findings suggest that the interdisciplinary approach led by nurses is well received by patients and has the potential to improve the quality and efficiency of care delivery for individuals with inflammatory rheumatological conditions in PC settings [101].

Riley et al. explored the contributions of nurse practitioners (NPs) in delivering rheumatology care in PC settings. NPs bring a unique skill set and perspective to the healthcare team, being involved in various aspects of patient care, e.g., performing physical assessments, ordering diagnostic tests, prescribing medications, providing patient education, and coordinating care with other healthcare providers. The survey highlighted the increasing demand for rheumatology services and the role of nurse practitioners in meeting this demand. As the prevalence of rheumatic diseases continues to rise, there is a growing need for healthcare professionals who can provide timely and accessible care to patients. NPs play a crucial role in expanding access to rheumatology services, particularly in underserved areas or where there is a shortage of rheumatologists [151].

Overall, the articles examined underscore the importance of interdisciplinarity in delivering rheumatology care and services, with nurse practitioners playing a central role in the healthcare team. By collaborating with rheumatologists and other healthcare professionals, nurse practitioners contribute to the delivery of high-quality, patient-centered care, ultimately improving outcomes for patients with rheumatic diseases.

#### 4.2.8. Digital Health (eHealth) Solutions

The electronic health record (EHR), which was first used in the UK, is a digital document that summarizes all data on the clinical history of the individual patient (visits, tests) and ensures the continuity and consistency of the diagnostic–therapeutic pathway [197]. It also allows different specialists to share patient information in real time, reducing working times and improving the quality of care. It is well known that the greatest use of EHRs takes place in PC [198]. In addition, EHRs are an important source of data for clinical and epidemiological research in everyday life [199]. It is possible to collect the data using structured codes or alternatively as free text. Numerous authors have proposed strategies to extrapolate the data contained in EHRs to obtain an early diagnosis of RA [200]. According to Nicholson A. et al., a diagnosis of RA could be made at least 6 months before its registration in the EHR using a structured diagnostic code. The authors proposed a strategy to create lists of marker codes for RA diagnosis combining a priori and a posteriori phases based on the data [201]. According to Ford E. et al., EHRs are a basic tool for estimating the incidence and early diagnosis of RA, provided that the data reported by GPs in free text form are also evaluated. Therefore, it is necessary to introduce IT methods that enable the automatic processing of information entered in free text, such as developing groups of indicator codes and searching for keywords, timing keywords in relation to codes, associating keywords with codes, comparing information in codes and keywords, and combining codes and keywords as predictors for case definition [152]. Zhou SM et al. developed an algorithm for RA phenotyping using data from EHRs. Using a machine learning approach, the authors identified predictive factors associated with diagnostic codes for RA, e.g., for DMARDs, and the absence of alternative diagnoses such as PsA [153].

EHRs can also be used to identify and quantify the risk of RA comorbidities. Ursum J. et al. found that 56% of adult patients had at least one comorbidity over a 10-year period after the onset of RA, most commonly cardiovascular (23%), MSK (17%), and neurological (8%) diseases. The highest hazard ratios were associated with anemia (HR 2.0), osteoporosis (HR 1.9), and chronic obstructive pulmonary disease (HR 1.8) [154].

eHealth tools can support GPs in different phases of the RA diagnostic–therapeutic process, contributing to reduced health costs [56,202]. The diagnosis step can be streamlined using digital diagnostic decision support tools, such as web tests and smartphone applications. The group of Knevel R. et al. developed and validated “Rheumatic?”, a web-based, patient-centered, multilingual diagnostic tool that enables the calculation of a risk score for arthritis at the exordium, including RA, and helps primary care physicians to differentiate it from nonrheumatic MSK diseases. Patients who reach a threshold of 2 need to be evaluated by a rheumatologist [155]. Reed M. et al. combined machine learning algorithms by developing a smartphone application. Each patient uploaded photos of their hands to the application and answered a 9-point questionnaire. The accuracy, precision, recall, and specificity in diagnosing RA were 85.1, 80.0, 88.1, and 82.7%, respectively [203]. This case is an example of the application of artificial intelligence in medicine.

The use of eHealth solutions enables patients to improve their self-management [204]. In an RCT, Rodríguez Sánchez-Laulhé P. et al. developed “CareHand”, a digital app with customized home exercises and tools to educate and inform RA patients. Compared to patients treated with the classical approach, RA patients using the CareHand application reported better outcomes in terms of overall hand function, work performance, pain, and satisfaction after 3 and 6 months [156].

Another potential of eHealth tools is the reduction in waiting times for doctor’s visits by allowing GPs to interact with rheumatologists via electronic consultations or structured group teleconferences [157]. The “eConsult Champlain BASE” (Building Access to Specialists Through eConsultation) service is a web-based application that allows patients to avoid in-person specialist visits or temporarily support them while they wait for an in-person consultation. GPs’ reports of RA are among the most common. The average time it took GPs to receive a response from the rheumatologist via the eConsult service was just 1 day, compared to a period of up to 155 days for routine specialist visits for RA. The level of satisfaction with eConsult was high among both GPs and patients [158]. According to Keely E. et al., the most important gaps in the management of rheumatic diseases can be derived from the questions GPs most frequently ask rheumatologists via eConsult. For RA, GPs’ main doubts concern the differential diagnosis between inflammatory and non-inflammatory pain and the interpretation of abnormal laboratory tests, such as RF, in the absence of signs and symptoms associated with rheumatic disease. Teleconferencing also allows for immediate interaction between rheumatologists and GPs [205]. One of the applications could be the screening of cardiovascular risk factors, as reported by Navarro-Millán I. et al., who focused on the screening and treatment of hyperlipemia in RA patients. The authors noted that both GPs and rheumatologists agree that screening for RA risk factors should be the prerogative of GPs. The most common barriers to the screening of hyperlipidemia perceived by rheumatologists are the lack of time and the actual effectiveness of screening. Conversely, GPs focused mainly on the possible occurrence of myalgia due to statins, on the side effects of multiple therapies and on the greater adherence to antirheumatic therapies rather than to statins. Both health operators emphasized the lack of knowledge of guidelines [159].

#### 4.2.9. Budget Impact Analysis

RA is one of the chronic diseases with the greatest economic impact worldwide [23]. The mean estimated annual cost for RA patients in Europe was EUR 12,902, while the total mean annual cost of the disease across all of Europe was estimated at about EUR 25.1 billion [206]. Analyzing the impact on the budget makes it possible to rescale diagnostic–therapeutic pathways to contain healthcare costs [207]. According to Kelleher D. et al., the application of a model for early detection and referral of RA patients in the Irish healthcare system allows the identification of RA patients within 3 months from the onset. In fact, it is notable that the earlier treatment is undertaken, the less joint damage and associated disability will develop. The criteria useful for GPs to refer a patient with suspected RA to a specialist include the presence of painful and swollen joints, a family history, morning stiffness, extra-articular features, an autoimmune panel, and inflammatory indices. The cost savings to the Irish healthcare system amount to EUR 237,547 over 5 years and are related to the reduction in the number of GP visits and the reduction in laboratory tests prescribed by GPs. The model helps to diagnose up to 750% more RA patients [160].

The main limitations of this study are the following: the different types of works examined with heterogeneous methods and disciplines (e.g., patient interviews or CAs), the diversity of interventions carried out in PC settings, the scarcity of RCTs, and the absence of a homogeneous control group.

## 5. Conclusions

Although new DMARDs can interrupt the clinical evolution of RA, early diagnosis and prevention of CVD in PC remain a challenge for healthcare systems. It is necessary to improve communication between healthcare professionals and patients so that GPs act as an intermediary between the patient and the specialist. Effective management of RA in PC involves the synergy of different healthcare specialists, in particular nurses and physiotherapists. Another critical issue identified is the poor training of both GPs and patients. The former should implement the ability to make early diagnosis, and the latter should be encouraged to self-manage chronicity. The application of eHealth solutions to PC allows us to overcome most critical issues related to RA management, containing healthcare costs and ensuring high patient satisfaction. Implementing these interventions in European PC units represents an excellent opportunity to improve the overall care of RA patients and effectively manage their comorbidities. This patient-centered approach, integrated with an interdisciplinary perspective, can help optimize the effectiveness and efficiency of RA management, improving patients’ quality of life and alleviating the burden on the healthcare system.

For this purpose, a change in the approach to RA by GPs and a reallocation of healthcare resources towards PC is mandatory. We hope that the approach to RA in PC could be configured as a reference model for the management of other chronic pathologies to which a high level of disability is related.

## Figures and Tables

**Figure 1 ijerph-21-00662-f001:**
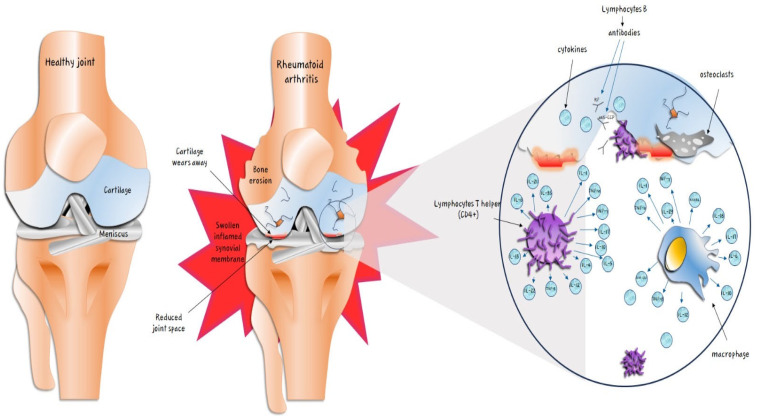
Pathogenetic mechanisms of rheumatoid arthritis (RA). Release of cytokines by T helper and B lymphocytes and macrophages. Macrophages appear to produce different interleukins (ILs), namely interleukin-1 (IL-1), interleukin-6 (IL-6), interleukin-10 (IL-10), interleukin-12 (IL-12), interleukin-17 (IL-17), interleukin-18 (IL-18), interleukin-23 (IL-23), interferon-gamma (INF-)γ, tumor necrosis factor-alfa (TNF-α), transforming grow factor-beta (TNF-β), granulocyte–macrophage colony-stimulating factor (GM-CSF), receptor activator of nuclear factor kappa-B ligand (RANKL). CD4+ T cells can release some of the same cytokines, namely IL-1, IL-6, IL-10, IL-12, IL-17, IFN-γ, TNF-α, and TGF-β. In addition, they can produce interleukin-4 (IL-4), interleukin-5 (IL-5), interleukin-13 (IL-13), interleukin-21 (IL-21), interleukin-22 (IL-22), and interleukin-35 (IL-35).

**Figure 2 ijerph-21-00662-f002:**
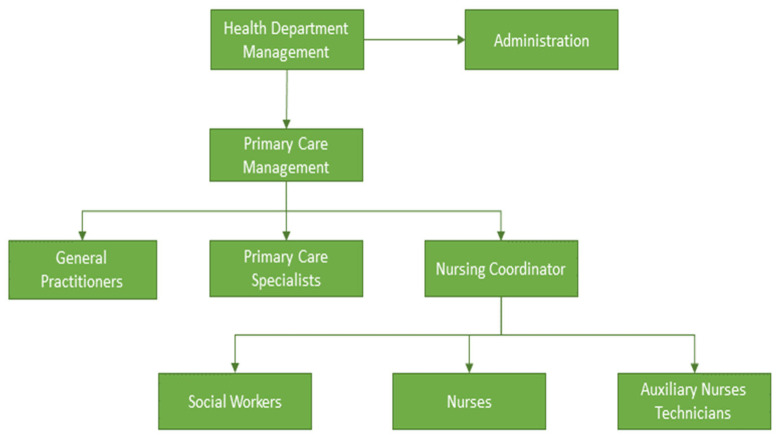
Flow-chart of primary care (PC) organization.

**Figure 3 ijerph-21-00662-f003:**
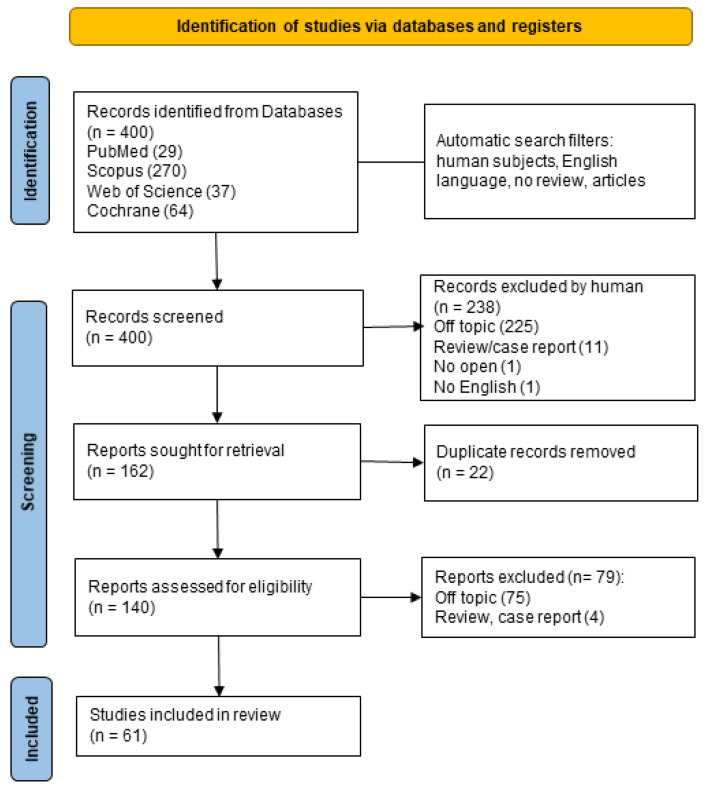
Preferred Reporting Items for Systematic Reviews and Meta-Analyses statement for reporting scoping reviews (PRISMA-ScR) flow-chart [86].

**Figure 4 ijerph-21-00662-f004:**
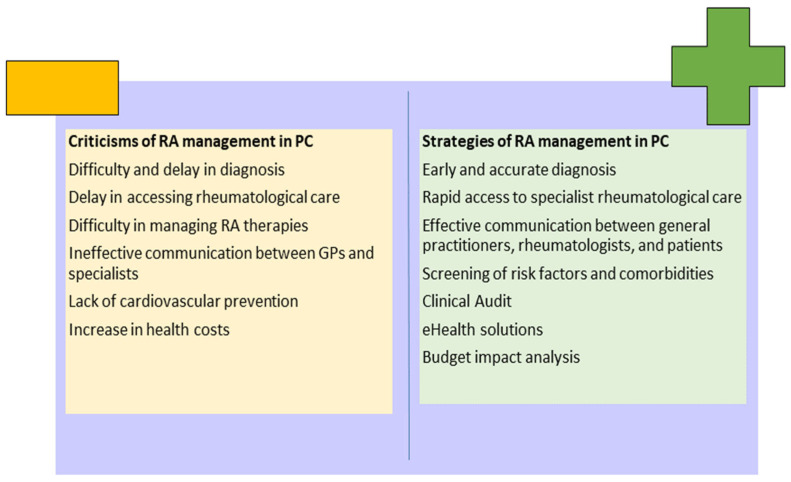
Criticisms and strategies of rheumatoid arthritis (RA) management in primary care (PC).

**Table 1 ijerph-21-00662-t001:** Article screening strategy.

Article screening strategy	KEYWORDS: A: “rheumatoid arthritis”; B: “primary care”
	Boolean Indicators: “A” AND “B”
	Timespan: from 1 January 2013 to 30 October 2023
	Electronic Databases: PubMed, Scopus, Web of Science, and Cochrane

**Table 2 ijerph-21-00662-t002:** Presentation of the extracted data. APC: anti-citrullinated protein antibody; axSpA: axial spondyloarthritis; BASE: Building Access to Specialists Through eConsultation; CC: case–control; CS: cross-sectional; CVD: cardiovascular disease; EHR: electronic health record; HR: hazard ratio; I: interview; IA: inflammatory arthritis; GCs: glucocorticoids; GPs: general practitioners; NSAIDs: non-steroidal anti-inflammatory drugs; OB: observational; PC: primary care; PsA: psoriatic arthritis; RCT: randomized controlled trial; RC: retrospective cohort; R: retrospective; RA: rheumatoid arthritis; RF: rheumatoid factor; UK: United Kingdom; USA: United States of America.

Authors	Study Sample and Country	Type of Study	Topic	Results
van Beers-Tas et al., 2020 [100]	2314	CC	Detection of crucial temporal patterns in PC for individuals later diagnosed with IA.	Marked rise in visits for musculoskeletal symptoms and infections almost one and a half years prior to the official diagnosis
Hider et al., 2021 [101]	4549	RCT	Detection of specific symptoms of IA in PC Interdisciplinary management of RA	54.3% of patients reported non-specific musculoskeletal problems, such as joint pain and stiffness. The study proposes a nurse-led model that integrates care across various healthcare disciplines
Newsum et al., 2016 [102]	126	RC	Detection of specific symptoms of IA in PC	Specific symptoms of IA, such as morning stiffness, compression testing, and family history, were rarely recorded by GPs
Morsley et al., 2019 [103]	495,434	RC	Incorrect use of diagnostic tests by GPs	Only 2.6% of RF-positive cases were diagnosed with RA within a year. With a sensitivity of 70.8%, a specificity of 78.7%, and a substantial cost of approximately EUR 3,963,472, equivalent to EUR 1432 per true positive, more precise patient selection is necessary for RF testing to reduce costs in PC
Singh et al., 2019 [104]	173	RC	Incorrect use of diagnostic tests by GPs	ACPA and RF were prescribed only in 34.45% of cases
Miller et al., 2015 [105]	62,436 UK	RC	Incorrect use of diagnostic tests by GPs	Among 62,436 RF tests, 7.5% were positive, with 57.8% of incident RA cases being seropositive. RF exhibited a positive likelihood ratio of 9.5 and a negative likelihood ratio of 0.5, with a sensitivity and specificity of 57.8% and 93.9%, respectively. Median time to referral after a positive RF test was significantly shorter than for negative tests (54 vs. 150 days)
Saad et al., 2020 [106]	66	RC	Delay in referral of RA patients to rheumatologists	Need for GPs to apply guidelines, such as EULAR 2016 for RA
Ledingham et al., 2017 [107]	6354 England, Wales	I	Delay in referral of RA patients to rheumatologists	Considerable gap between the standards and the delivery of care, subject to geographic variability, according to NICE guidelines
Stack et al., 2019 [108]	822 UK	OB	Delay in referral of RA patients to rheumatologists	Median time between symptom onset and seeing a rheumatologist is 27.2 weeks
Widdifield et al., 2014 [109]	19,760 Canada	RC	Delay in referral of RA patients to rheumatologists	The disparities are influenced by external factors such as proximity to rheumatologists, socioeconomic status, and male physician sex
Badley et al., 2015 [110]	105 health planning areas Canada	R	Delay in referral of RA patients to rheumatologists	Identification of barriers to rheumatological care particularly for populations with low access GPs
Chilton et al., 2021 [111]	11 England	I	Delay in referral of RA patients to rheumatologists	The delay in referral to specialist care is a source of great dissatisfaction for RA patients and their families, generating frustration
Laires et al., 2013 [112]	34 Portugal	I	Delay in referral of RA patients to rheumatologists Communication between healthcare operators	Identified barriers are the following: difficulty in accessing PC services, delays in diagnosis by GPs, inefficiencies in referrals to specialized care, and controlled processes in prescribing biologic drugs in public hospitals. Proposed solutions include improving epidemiological and clinical knowledge of RA, enhancing disease understanding among patients and GPs, and increasing awareness of current treatment guidelines
Lee et al., 2021 [113]	120 rheumatology eConsults	Descriptive CS	Difficulty in the management of RA treatment	Rheumatologists and GPs use DMARDs only in 56% and 30% of patients, respectively. Only 74% of patients are visited by a rheumatologist. High prescription rate of GCs.
Black RJ et al., 2015 [114]	7777 UK	R	Difficulty in the management of RA treatment	50% of patients initially diagnosed with RA took GCs prescribed by GPs and had advanced age and CVD
Widdifield J. et al., 2017 [115]	2430 Canada	OB	Difficulty in the management of RA treatment	NSAIDs and GCs the drugs most ordered by GPs
De Barros Lopes et al., 2021 [116]	50	I	Difficulty in the management of RA treatment	Patients taking MTX are subjected to laboratory monitoring more frequently than 3 months, as recommended by BSR
Wong et al., 2019 [117]	2430 Canada	R	Ineffective communication between RA healthcare professionals	Referral letters by rheumatologists are deficient in critical information
Barber et al., 2016 [118]	170 charts	R	Lack of CVD preventive care/communication between healthcare operators	The prevalent CVD risk factors for RA are hypertension, obesity, smoking, and dyslipidemia. There are 11 quality indicators for CVD management in RA patients, such as inadequate documentation of a formal cardiovascular risk assessment, failure to communicate increased cardiovascular risk to GPs, and a lack of discussion on the risks and benefits of anti-inflammatory agents
Bartels et al., 2016 [119]	31	I	Lack of CVD preventive care	Preventive treatment is typically inadequate both by GPs and rheumatologists, despite the elevated risk of CVD
Bartels et al., 2014 [120]	14,974 USA	Cohort	Lack of CVD preventive care	Among patients with undiagnosed hypertension, those with RA had a lower probability (36%) of receiving a diagnosis of hypertension than those without RA (51%)
Nikiphorou et al., 2015 [121]	101 Norfolk	I	Increased health costs for RA	Substantial increase in direct medical costs over a 6-month period, primarily driven by using biologic DMARDs for the RA subgroup (51% of total costs)
Scott et al., 2018 [122]	1388 GPs UK	CS survey	Early and accurate diagnosis of RA	Improvement of clinical history and examination findings in the early diagnosis of RA, especially small joint swelling and pain. Paradigm shift in GPs’ approaches, prioritizing clinical findings over investigations to meet referral timelines
Van Dijk B.T. et al., 2020 [123]	997 patients The Netherlands	CS	Early and accurate diagnosis of RA	A simple question about difficulties with dressing could aid in early IA detection
Garcia-Montoya, L., 2022 [124]	6780 individuals with non-specific MSK symptoms UK	OB	Early and accurate diagnosis of RA	Identification of specific predictors of IA, such as high APCA levels and pain in hands or feet
Nam J. et al., 2016 [125]	2028 individuals with non-specific MSK symptoms UK	Prospective cohort	Early and accurate diagnosis of RA	Patients with non-specific MSK symptoms without clinical synovitis and APCA positivity have a high risk of rapidly developing RA
Almoallim H., 2017 [126]	203 IA patients Saudi Arabia	CS	Early and accurate diagnosis of RA	The authors identified nine variables for early referral with high specificity and good predictive value, including loss of appetite, joint swelling, and positive tests for RA and APCA antibodies
Van Delft Etam et al., 2022 [127]	250 patients, of whom 42 with IA Netherlands	Prospective OB	Early and accurate diagnosis of RA	The CARE model is the superior model for early referral. Sex, age, joint features, and duration of symptoms are important for early diagnosis
Ten Brinck RM et al., 2018 [128]	644 patients Netherlands	R	Early and accurate diagnosis of RA	Identification of a method for GPs in detecting synovitis through joint examination
Nielen M. et al., 2013 [129]	219 IA patients Netherlands	R	Early and accurate diagnosis of RA	An algorithm based on EMR, incorporating additional diagnostic information, increases the percentage of correctly diagnosed IA patients to 78%
Muller S. et al., 2015 [130]	4161 people with a first read code for RA UK	R	Early and accurate diagnosis of RA	An algorithm based on CPRD, reflecting changes in coding practices and the introduction of new treatments, revealed an increased rate of confirmed RA diagnoses (86%)
Cummins et al., 2015 [131]	457 referrals Australia	R	Early and accurate diagnosis of RA	The application of the 2010 ACR/EULAR criteria for the early diagnosis of RA in PC improves triage decisions and reduces waiting times
Kvien et al., 2020 [132]	180 RA patients Europe	I	Rapid access to specialist rheumatological care Effective communication between healthcare operators	The authors suggested 18 best-practice interventions, e.g., support for self-management, early arthritis clinics, rapid access to care through online referral, triage, and ultrasound-guided diagnosis
Russell-Westhead M., 2010 [133]	3363 arthritis patients UK	OB	Effective communication between healthcare operators	Group consultations can be sustainable, clinically effective, and efficient for the monthly review of active disease and annual review of stable disease. Key success factors include efficiency, empathy, education, patient involvement, and autonomy
Magliah M. et al., 2019 [134]	203 patients 30 GPs 3 rheumatologists Saudi Arabia	R	Effective communication between healthcare operators	Standardized training of GPs conducted by rheumatologists could improve the accuracy of early diagnosis of IA
Fonseca et al., 2018 [135]	144 in the referral support actions group 196 in the control group Portugal	RCT	Effective communication between healthcare operators	A referral support program is effective for identifying patients with RA and axSpA
Movahedi M. et al., 2016 [136]	21,962 patients UK	Cohort	Screening of risk factors and comorbidities	The use of GCs represents a significant independent risk factor for the development of DM in RA patients, highlighting the importance of dosage and treatment duration. The HRs were 1.30 (CPRD) and 1.61 (NDB)
Stouten et al., 2021 [137]	38 RA, 229 SpA, 167 PsA patients Belgium	CC	Screening of risk factors and comorbidities	Depression, cardiovascular diseases, and opioid use were particularly high in patients with RA and PsA
Monk et al., 2013 [138]	401 RA patients 1198 controls UK	R, CC	Screening of risk factors and comorbidities	Additional screening for CVD is necessary in PC settings
Nikiphorou et al., 2020 [139]	6591 RA patients	R, CC	Screening of risk factors and comorbidities	Identifying an excess of stroke and heart failure before the diagnosis of RA, the study highlighted an increased risk of CVD post-diagnosis
Joseph et al., 2016 [140]	5677 patients UK	R, cohort	Screening of risk factors and comorbidities	In RA patients, smoking was associated with an increased risk of all-cause mortality, CVD, and respiratory tract infections
Joseph et al., 2017 [141]	5677 patients UK	R, cohort	Screening of risk factors and comorbidities	Each year of smoking cessation was associated with a reduced risk of mortality for RA patients
Abhishek et al., 2018 [142]	21,622 RA patients 86,488 controls UK	CC	Screening of risk factors and comorbidities	From 2005 to 2009, the RA mortality rate decreased by 7.7% annually, while controls improved by 2.2% annually from 1990 to 2009. Despite a 32% excess mortality risk in the entire cohort, cases incident after 2006 exhibited a 15% decrease. The authors observed that improvement in RA management may contribute to reduced RA mortality
Pratt A.G. et al., 2016 [143]	176 RA patients in England	OB, R	Clinical audit	The PC physicians referred only 17% of patients within 3 days of the first presentation, and only 38% received a specialist evaluation within 3 weeks of referral. Only 53% of RA patients were treated with DMARDs within 6 weeks of referral
Stack R.J. et al., 2014 [144]	19 GPs UK	I	Clinical audit	Delays in presentation to GPs for patients with specific autoantibodies
Persell et al., 2020 [145]	317 RA patients USA	I	Active RA patient involvement	During the six-month intervention period, an increase in the use of statins for primary prevention was observed, rising from 18.4% to 23.8%, while diagnoses and control of hypertension significantly improved
Curtis et al., 2021 [146]	307 subjects UK	R, CS	Active RA patient involvement	The Methotrexate Experience Questionnaire is a reliable and valid tool for assessing the adherence of RA patients to MTX treatment
Lopatina et al., 2019 [147]	38 key RA stakeholders (patients, healthcare professionals, healthcare administrators, researchers) Canada	Case study	Active RA patient involvement	Involvement of RA patients in activities such as project vision development, prospective data collection through patient-to-patient studies, and definition of future evaluation strategies
Puchner et al., 2017 [148]	5294 people with undifferentiated arthritis, of which 1765 had RA Austria	I	Interdisciplinary management of RA	Involving and coordinating various healthcare professionals, including rheumatologists, GPs, physiotherapists, nurses, and other specialists
Bakker et al., 2023 [149]	140 RA or axial SpA patients The Netherlands	RCT	Interdisciplinary management of RA	Physiotherapists provide exercise therapy, occupational therapy, and psychosocial support
Ndosi et al., 2014 [150]	91 RA patients	RCT	Interdisciplinary management of RA	Non-inferiority of RA care provided by nurses working in PC vs. physicians
Riley et al., 2017 [151]	3455 nurse practitioners USA	I	Interdisciplinary management of RA	Nurse practitioners bring a unique skill set, performing physical assessments, ordering diagnostic tests, prescribing medications, providing patient education, and coordinating care
Ford E. et al., 2013 [152]	6387 patients UK	R	Digital health solutions	It is necessary to introduce IT methods in EHRs that enable the automatic processing of information entered in free text
Zhou SM et al., 2016 [153]	2,238,360 patients UK	R	Digital health solutions	Using a machine learning-based algorithm, the authors identified predictive factors associated with diagnostic codes for RA in EHR
Ursum J. et al., 2013 [154]	3354 patients with newly diagnosed IA The Netherlands	CC	Digital health solutions	Use of EHR to identify comorbidities associated with RA
Knevel R. et al., 2022 [155]	50 individuals with MSK symptoms and APCA 52 patients with early joint swelling Sweden	R, cohort	Digital health solutions	Development of “Rheumatic?”, a web-based multilingual diagnostic tool that enables the calculation of a risk score for arthritis at the exordium, including RA, and helps in differential diagnosis
Rodríguez Sánchez-Laulhé P., 2022 [156]	36 RA patients Spain	RCT	Digital health solutions	Development of “CareHand”, a digital app with customized home exercises and tools to educate and inform RA patients
Pfeil, J.N., 2023 [157]	50,185 patients, 50,124 controls Brazil	R, cohort	Digital health solutions	Development of “eConsult Champlain BASE” service, a web-based application that allows patients to avoid in-person specialist visits
Keely, E.; 2021 [158]	300 consecutive faxed referrals and 300 eConsult referrals Canada	Comparative	Digital health solutions	eConsult is useful to ease GPs’ main doubts about RA, e.g., the differential diagnosis between inflammatory and non-inflammatory pain and the interpretation of abnormal laboratory tests
Navarro-Millán I., 2020 [159]	27 rheumatologists	I	Digital health solutions	Screening of cardiovascular risk factors through teleconferences, such as treatment of hyperlipemia in RA patients
Kelleher D. et al., 2020 [160]	4,857,000 Irish population	Bias model based on incidence	Budget impact analysis	The application of a model for the early detection and referral of RA patients allows the identification of RA patients within 3 months from the onset, reducing healthcare costs

## Data Availability

Not applicable.

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
