# Peer review of "Management of Rheumatoid Arthritis in Primary Care: A Scoping Review"

_ijerph, 2024, doi:10.3390/ijerph21060662_

Round 1

Reviewer 1 Report

Comments and Suggestions for Authors

The authors wrote a scoping review of the management of rheumatoid arthritis in primary care.

Although this research is not novel, it may require revisions and/or clarifications before it can be considered for publication.

Abstract

- Briefly give objectives, sources of evidence, and charting methods.

Introduction

- You may highlight the epidemiology of RA, including its prevalence and female-to-male ratio.

DOI: 10.1177/03000605231204477

- You may draw attention to the RA patient's annual healthcare costs.

DOI: 10.3389/fmed.2023.1221393

- The gaps in knowledge and rationale for the study need to be mentioned.

- One such angle to address is the equilibrium between the availability and demand for rheumatology care.

 DOI: 10.3389/fmed.2020.00016

Methods

- Please outline this section following the (PRISMA-ScR) Checklist guidelines.

- Selection of sources of evidence: State the process for selecting sources of evidence (i.e., screening and eligibility) included in the scoping review.

- Data items: List and define all variables for which data were sought and any assumptions and simplifications made.

- Synthesis of results: Describe the methods of handling and summarizing the data that were charted.

Results

-  For each source of evidence, present characteristics for which data were charted and provide the citations. 

References
Numerous references require updating. Verify and make the necessary corrections. Additionally, try to use references that are as current as possible; stay away from references that are older than five to ten years. Throughout your essay, include suggested recent references and, if possible, remove outdated and outdated references.

Author Response

REVIEWER 1

The authors wrote a scoping review of the management of rheumatoid arthritis in primary care.

Although this research is not novel, it may require revisions and/or clarifications before it can be considered for publication.

Abstract

- Briefly give objectives, sources of evidence, and charting methods.

The suggested indications were inserted from line 22 to line 26 of the abstract.

Introduction

- You may highlight the epidemiology of RA, including its prevalence and female-to-male ratio.

DOI: 10.1177/03000605231204477

- You may draw attention to the RA patient's annual healthcare costs.

DOI: 10.3389/fmed.2023.1221393

- The gaps in knowledge and rationale for the study need to be mentioned.

- One such angle to address is the equilibrium between the availability and demand for rheumatology care.

 DOI: 10.3389/fmed.2020.00016

We thank and welcome the reviewer's suggestions, which we have included in the introduction (lines 46, 46, 75, 76, 99-103).

Methods

- Please outline this section following the (PRISMA-ScR) Checklist guidelines.

- Selection of sources of evidence: State the process for selecting sources of evidence (i.e., screening and eligibility) included in the scoping review.

- Data items: List and define all variables for which data were sought and any assumptions and simplifications made.

- Synthesis of results: Describe the methods of handling and summarizing the data that were charted.

Thank you. We made the changes indicated in the materials and methods paragraph (subparagraphs 2.3, 2.4, 2.5)

Results

-  For each source of evidence, present characteristics for which data were charted and provide the citations.

We are grateful for this opportunity for improvement. The data sources have been summarized in Table 2.

References
Numerous references require updating. Verify and make the necessary corrections. Additionally, try to use references that are as current as possible; stay away from references that are older than five to ten years. Throughout your essay, include suggested recent references and, if possible, remove outdated and outdated references.

References 89, 158 and 160 have been eliminated because they are obsolete. The other references have a year of publication no older than 2013.

Reviewer 2 Report

Comments and Suggestions for Authors

The Manuscript: "Management of Rheumatoid Arthritis in Primary Care: a scoping review" by Francesco Inchingolo and colleagues delves into the role of general practitioners in primary care in the management of rheumatoid arthritis, addressing various factors such as the time of disease diagnosis, accessibility to specific treatments (DMARDs), or healthcare costs. Having carefully reviewed the manuscript, I have a few comments to share.

  In order to improve the understanding of the role of different cells in rheumatoid arthritis, Figure 1 should depict cytokines being released by macrophages and antibodies by B lymphocytes. The presence of helper T lymphocytes (CD4+) should also be shown. Additionally, in that figure, osteoclasts should be placed in the bone area.   In relation to the previous observation, the presence of the different cells and molecules involved in the pathogenesis of the disease should be mentioned in the text of the manuscript, even if briefly.   In line 234, the acronym IA is mentioned, but it is not defined as inflammatory arthritis until line 360. All acronyms should be defined the first time they appear. Comments on the Quality of English Language  

Minor errors have been detected

Author Response

REVIEWER 2

The Manuscript: "Management of Rheumatoid Arthritis in Primary Care: a scoping review" by Francesco Inchingolo and colleagues delves into the role of general practitioners in primary care in the management of rheumatoid arthritis, addressing various factors such as the time of disease diagnosis, accessibility to specific treatments (DMARDs), or healthcare costs. Having carefully reviewed the manuscript, I have a few comments to share.

In order to improve the understanding of the role of different cells in rheumatoid arthritis, Figure 1 should depict cytokines being released by macrophages and antibodies by B lymphocytes. The presence of helper T lymphocytes (CD4+) should also be shown. Additionally, in that figure, osteoclasts should be placed in the bone area.   In relation to the previous observation, the presence of the different cells and molecules involved in the pathogenesis of the disease should be mentioned in the text of the manuscript, even if briefly.  

We are grateful for this opportunity for improvement. In the introduction (lines 48-53) a brief summary of the pathogenesis of rheumatoid arthritis was added, relating it to Figure 1, to which a more detailed caption was added (lines 60-70).

In line 234, the acronym IA is mentioned, but it is not defined as inflammatory arthritis until line 360. All acronyms should be defined the first time they appear.

The acronym "IA" was made explicit on line 234, the term "inflammatory arthritis" was eliminated on line 360.

Comments on the Quality of English Language

Minor errors have been detected.

We have corrected several inaccuracies in the text, highlighting them in purple.

Reviewer 3 Report

Comments and Suggestions for Authors

The review by Inchingolo and colleagues provides a comprehensive overview of the management of RA in primary care following the guidelines of meta-analyses statement for reporting in a scoping review format.

The review is insightful and addresses key gaps in quality improvement of clinical practice. The only issue that I would like the authors to address in more detail would be a discussion of regional differences in the approach to RA by general practitioners and a reflection of the distribution of healthcare resources towards primary care in RA diagnosis and management in different health care systems.

Comments on the Quality of English Language

There are a few minor issues, for example the use of 'citokine' in Fig. 1.

Author Response

REVIEWER 3

The review by Inchingolo and colleagues provides a comprehensive overview of the management of RA in primary care following the guidelines of meta-analyses statement for reporting in a scoping review format.

 The review is insightful and addresses key gaps in quality improvement of clinical practice. The only issue that I would like the authors to address in more detail would be a discussion of regional differences in the approach to RA by general practitioners and a reflection of the distribution of healthcare resources towards primary care in RA diagnosis and management in different health care systems.

We are grateful for the suggestions, which have been included in the introduction from lines 104 to 123 with their references.

Comments on the Quality of English Language

There are a few minor issues, for example the use of 'citokine' in Fig. 1.

Figure 1 has been modified to read “cytokine”. Other inaccuracies have been corrected in the text by highlighting them in purple.

Round 2

Reviewer 1 Report

Comments and Suggestions for Authors

The authors wrote a scoping review of the management of rheumatoid arthritis in primary care.

This revised version of the manuscript has substantially improved, and the authors have addressed all the comments successfully.